# Stabilization of Chromium Waste by Solidification into Cement Composites

**DOI:** 10.3390/ma16186295

**Published:** 2023-09-20

**Authors:** Cherif Belebchouche, Salah-Eddine Bensebti, Chaima Ould-Said, Karim Moussaceb, Slawomir Czarnecki, Lukasz Sadowski

**Affiliations:** 1Department of Civil Engineering, Faculty of Sciences of Technology, Frères Mentouri Constantine 1 University, Constantine 25000, Algeria; s_bensebti@yahoo.fr; 2Materials and Durability of Constructions Laboratory, Faculty of Sciences of Technology, Frères Mentouri Constantine 1 University, Constantine 25000, Algeria; 3Laboratory of Materials Technology and Process Engineering, Faculty of Technology, University of Bejaia, Bejaia 06000, Algeria; 4Laboratory of Biology and Physiology of Organisms (LBPO), Faculty of Biological Sciences, USTHB, BP 32 El-Alia, Bab Ezzouar 16111, Algeria; ouldsaidchaima93@gmail.com; 5Department of Materials Engineering and Construction Processes, Wroclaw University of Science and Technology, Wybrzeze Wyspianskiego 27, 50-370 Wroclaw, Poland; lukasz.sadowski@pwr.edu.pl

**Keywords:** hazardous chromium wastes, acidic environment, cementitious materials, leaching

## Abstract

This article deals with the study of hazardous chromium leaching, stabilized/solidified by cement CEM II after 28 days of curing, in an acidic environment. The mortars subjected to this study were investigated by X-ray diffraction (XRD) characterization to evaluate the influence of chromium waste on their mineralogical structure. In the study range (0.6–1.2%), increasing the mass percentage of Cr_2_O_3_ in the mortars indicates that chromium accelerates the hydration process and setting of the mortar and increases the mechanical strength of the mortars compared to the control sample. It was observed that the release of chromium during the Toxicity Characteristic Leaching Procedure (TCLP) test and the efficiency of the stabilization/solidification process depended on the initial Cr concentration and the leaching time. The use of XRD allowed the identification of new crystallized phases in the cement matrices, namely, CaCrO_4_·2H_2_O and chromium–ettringite Ca_6_Cr_2_(SO_4_)_3_(OH)_12_·26H_2_O, which confirms the immobilization of chromium and the efficiency of the stabilization/solidification process. In this research, the release mechanism was found to be primarily a surface phenomenon by modeling the experimental data (dissolution or precipitation).

## 1. Introduction

Currently, more attention is being paid to protecting human health and the environment from the dangers of hazardous waste. One of these forms of hazardous waste is chromium obtained from the production of cement [1], ceramic, stainless steel, and other alloys from chromium plating [2,3,4]. Cr (VI) stands out among other chromium forms when considering toxicity, solubility, and mobility [5,6]. Chromium is very dangerous for both human health and the environment. When its concentration exceeds admissible values, it causes the following problems: rashes, heart problems, disturbances of metabolism and diabetes, upset stomach and ulcers, respiratory problems, weakened immune system, liver and kidney damage, alteration of genetic material, lung cancer, birth defects, infertility, formation of tumors, and death [7].

Therefore, the issue of environmental contamination by Cr (VI) is receiving increasing attention due to its wide distribution worldwide, particularly in water [8] and soil [9]. This contamination is a result of both natural processes and human activities [10], including mining, metal works, steel and metal alloy production, paint manufacturing, wood and paper processing, dyeing, and the discharge of chromium into wastewater [6,11,12]. Furthermore, the incineration of coal or municipal waste for energy generation and the production of second-generation fertilizers contribute to the elevated levels of Cr (VI) in soil and water through the deposition of ash [13]. Improper disposal of this heavy metal can lead to severe environmental and ecological problems, as water serves as the primary carrier of environmental pollutants. In Europe, the permissible concentration values for Cr (VI) range from 0.05 to 2 mg/L, as determined by the environmental policies of countries such as Norway, Poland (most precautionary value), and the Netherlands. In Algeria, the permissible concentration values for Cr (VI) and Cr (III) are 5 mg/kg and 50 mg/kg, respectively [14,15]. According to the latest statistical compendium on raw hides and skins published by the Food and Agriculture Organization of the United Nations (FAO) [16], the annual global production of bovine hides and skins amounts to approximately 6.5 million metric tons (wet salted weight), with over 4.3 million metric tons being processed in developing countries. Sheepskins, lambskins, goatskins, and kidskins are processed globally for approximately 750 thousand metric tons (dry weight). Therefore, based on rough estimates, the tanning industry generates approximately 3.5–4.0 million metric tons of solid waste annually, a significant portion of which (up to 30–35% [17]) may contain chromium.

All these threats have pushed researchers to find appropriate treatment solutions for these heavy metals [18]. In the case of chromium, the powerful techniques are vitrification [19,20] or solidification/stabilization (S/S). Solidification has especially shown its effectiveness in reducing the polluting character of several types of hazardous wastes (by rendering them inert), namely, heavy metal contaminated soil [21], marine sediment [22], industrial waste [23], municipal solid waste incineration fly ash [7,24], and Cr (III) and Cr (VI) from acid mine drainage (AMD) [25]. S/S treatment with cement is one of the most widely used techniques due to its (1) very low cost and its ease of implementation compared to the vitrification technique, which is commonly used for nuclear waste; (2) its ability to render these heavy metals from toxic contaminants into a solid; and (3) the low solubility of cement matrix with a reduced environmental impact [26].

Characterizing the products of stabilization/solidification by leaching tests and characterization methods (e.g., XRD) has been used by many researchers throughout the world. These scenarios have been proven to help determine these heavy metals’ released concentrations and crystallized phases [27,28]. Furthermore, the results of leaching tests and characterization methods can be successfully used to confirm the mechanisms and reactions of the immobilization of these metals within the cement matrix. Previous studies have been conducted using cement, bentonite, coal fly ash, and metallurgical slag-based cementitious material for the immobilization of Hg, Pb, As, Cr, and Cd [29,30,31], Pb and Zn [32], Ni [33], Cr VI [25], and Pb, Zn, Cr, and Hg [34].

At present, and despite the advances made in the field of characterization of cement pastes, researchers have been unable to identify and determine the morphology of hydrates. Taking into account that these hydrates are formed in the presence of heavy metals and cement, an understanding of the mechanisms of immobilization is made more difficult [35,36]. In addition, it is difficult to identify all the phases formed during the hydration of cement in the presence of chromium, to evaluate the degradation of cementitious materials containing chromium, and to model the release of chromium from cementitious matrices. Moreover, it is very rarely discussed in the literature. In this context, it would be advantageous to carry out a deep experimental study to understand the phenomena that govern the process of stabilization/solidification and the release of chromium from cementitious matrices.

Considering the above, the authors performed complex research to evaluate the possibility of solidification/stabilization of chromium by cement CEM II. This complex research contains multiscale analyses of chromium release from different cement matrices that vary in the percentages of mass of the pure waste (Cr_2_O_3_). These analyses were combined with statistical criteria using the empirical Cote model. Furthermore, this study enables an evaluation of the degradation of the stabilized/solidified mortars that are rarely studied to complete previous works in the field of the solidification/stabilization of chromium.

## 2. Materials and Methods

### 2.1. Materials and Preparation of Mortar Mixes

The constituents used in formulations of the synthesized materials are Portland cement (CEM II 42.5), from the Ain El Kebira cement factory in Algeria and synthetic waste composed of standard sand, from the New Littoral Company in France, while pure pollutant Cr_2_O_3_ (CAS: 1308-38-9) is introduced in the form of pure laboratory products, to facilitate the tracking and detection of this metal inside the cementitious matrix. Demineralized water was used as mixing water to avoid any possible contamination by trace elements. The chosen content of pollutant (Cr_2_O_3_) of 0.6%, 0.96%, and 1.2% (mass) for pollution levels is considered plausible [35]. The chemical composition of the synthesized materials developed is recorded in Table 1 and their grain size distributions are presented in Figure 1.

Each mortar is formulated using a 3 kg mixer. Initially, sand and the pollutant are blended at a moderate velocity to achieve a uniform mixture. Subsequently, cement and demineralized water are introduced, and the entire blend is mixed for a brief duration.

It is then necessary to stop the agitation to scrape the bottom of the container with a spatula so that the hydration occurs homogeneously. After mixing, the mortars are poured into 4 × 4 × 16 cm^3^ and 7 × 7 × 28 cm^3^ molds and stored airtight at room temperature (20 ± 3 °C) for 28 days.

The rapid formation of a thin layer of surface calcite CaCO_3_ [36] by atmospheric carbon dioxide can impede the surface porosity and have considerable repercussions on the leaching rate, making carbonation of the surface a significant issue during sample preparation. This leads us to obtain a sample preparation method designed to minimize exposure to atmospheric conditions. We employed a water-to-cement ratio of 0.5 to promote effective diffusion [37], allowing us to obtain a constant capacity without needing a “vibrating table”.

After 28 days of cure in a wet room, the specimens of dimension 4 × 4 × 16 cm^3^ underwent mechanical tests according to the standard CEN 196-1. Each sample was tested in triplicate. On the other hand, the 7 × 7 × 28 cm^3^ specimens were dry-cut to prepare 3 × 3 × 3 cm^3^ monoliths from their cores to run the TCLP test. XRD and infrared spectrometry (FTIR) were then used to characterize the synthesized materials at different times (before and after leaching). The diffractometer used in our study was of type X’Pert PRO PANalytical. The samples were subjected to analysis in their powdered form, with particle sizes smaller than 100 μm.

XRD patterns and FTIR spectra were treated using X’Pert High Score (version 2. 1b 2.1.2, 9 January 2005) and IR solution software (version 1.4, 12 October 2007), respectively.

### 2.2. Testing Methodology

#### 2.2.1. Toxicity Characteristic Leaching Procedure (TCLP) Test

The TCLP test aims to characterize the behavior of the stabilized/solidified waste over time. S/S materials are subjected to the standard TCLP protocol, except in this study, the test duration is over 50 h [38] instead of 18 h. The leachant is a solution that is a mixture of 63.4 mL of 1 N sodium hydroxide and 5.7 mL of 1 N acetic acid, adjusted with distilled water to a 1 L solution. The pH of the leaching solution is 4.93. The liquid-to-surface ratio (L/S) is maintained at 20 cm^3^/cm^2^ under leaching conditions. The mortars made up are crushed to a diameter less than 9.5 mm, then they are submitted in contact with the leachant in 1 L containers, which are tightly closed to avoid, on the one hand, the admission of air, thus the risk of carbonation, and, on the other hand, the evaporation of the solution. In addition, agitation is ensured during the entire test. During the TCLP test, six samples were taken after 2, 4, 18, 26, 42, and 50 h of leaching. They were filtered and acidified to pH = 2 with 68% nitric acid (HNO_3_). They were then stored in airtight vials at room temperature. It is important to highlight that prior to analysis using flame atomic absorption spectrophotometry (FAAS) with the Aurora Instruments AI 1200, the samples underwent centrifugation to convert the different chromium forms into total chromium.

To assess the degradation of S/S mortars, a solid block measuring 3 × 3 × 3 cm^3^ is placed in contact with a fixed volume of leachant. The leachant used corresponds to that employed in the Toxicity Characteristic Leaching Procedure (TCLP) test. The leachant is replaced at intervals of 0.25, 0.75, 1, 2, 5, 7, 20, and 28 days, resulting in a total leaching period of 64 days. This duration is equivalent to 96 years of continuous leaching under experimental conditions. During each leachant replacement, the liquid-to-surface ratio (L/S) is maintained at a constant value of 10 cm^3^/cm^2^, and the experiment is conducted at room temperature (23 ± 1 °C). The setup is shielded from light and protected against air penetration (CO_2_).

Upon completing the test, the monolithic blocks are subjected to X-ray diffraction (XRD) analysis to assess the extent of degradation in terms of peak intensities related to portlandite.

The experimental program undertaken in this study is summarized in Figure 2.

#### 2.2.2. Modeling and Simulation of Cr Leachate Release from Monoliths

By integrating the cumulative fraction expressions from three equations established in the Cote model [39,40,41], we can formulate a semi-empirical expression for the cumulative leached fractions of chromium in the following general format:(1)$$CLF(t)=\frac{M(t)}{M_{0}}=K_{1}(1-e^{-K_{2}\;t})+K_{3}\sqrt{t}+K_{4}\;t$$
where M(t) is the concentration of Cr in the leachate in (μmol/L); M_0_ is the initial concentration of Cr introduced into the sample in (μmol/L); and K_1_, K_2_, K_3_, and K_4_ are the parameters that were determined using the convergence of the Cote model by the simplex method (using ORIGIN 6.0 MicroCal Software) on the experimental data of the released chemical species.

The initial component of Equation (1) accounts for the influence of species whose release kinetics are dictated by interactions between the sample surface and the leachant. The subsequent term in Equation (1) represents the outcome of solving diffusion equations (Fick’s laws), illustrating the impact arising from the release of species controlled primarily by pure diffusional transport. Lastly, the third term, denoted as K_4_t, encapsulates the concept of species release governed by first-order dissolution kinetics.

Therefore, we shall conduct our predictive research utilizing the empirical model of Cote, integrating surface phenomena with diffusion and chemical reactions. We use four statistical criteria for validating the Cote model: the residual variance, the correlation coefficient, the Student’s test, and the Fisher–Snedecor test.

R^2^ is the proportion of the variance of y explained by the explanatory variables. It is used to test the goodness of fit of y by ŷ and is expressed by the following relationship:(2)$$R^{2}=\frac{\sum_{i=1}^{n}{\left(\hat{y_{i}}-\bar{y}\right)}^{2}}{\sum_{i=1}^{n}{\left(y_{i}-\bar{y}\right)}^{2}}$$

The following quantity gives the part not explained by the regression and is called residual variance:(3)$$S_{r}^{2}=\frac{1}{n}\sum_{i=1}^{n}{\left(y_{i}-\hat{y}\right)}^{2}$$

The estimation of the influence of the regression coefficients is performed by the Student’s *t*-test, which is based on statistics:(4)$$T=\frac{\left|C_{j}\right|}{\sigma (C_{j})}$$

However, the overall validity of the model is assessed using the Fisher test, which relies on the following statistical measure:(5)$$F=\frac{\sum_{i=1}^{n}{\left(y_{i}-\bar{y}\right)}^{2}/m-1}{\sum_{i=1}^{n}{\left(y_{i}-\hat{y}\right)}^{2}/n-m}=\frac{\frac{R^{2}}{m-1}}{\frac{1-R^{2}}{n-m}}$$

## 3. Results and Discussion

### 3.1. Mechanical Tests

The results of the compressive strength (Rc) and flexural strength (Rf) tests are presented in Figure 3a,b and Table 2.

The results obtained show that the mechanical resistance values recorded for the S/S materials comply with those required by the X31-211 standard (strengths greater than 1 MPa) [42].

It was observed that increasing the amount of chromium in the mixtures increases the mechanical strength of mortars. This is because chromium can precipitate in the form of oxides, sulfates, and carbonates, as well as a substitute for calcium. In addition, it gives rise to new compounds that accelerate the hydration of cement grains and the setting of the material. The rapid precipitation of C-S-H and portlandite due to the addition of chromium confers good mechanical strength to the hardened cementitious matrix. In summary, it can be concluded that the mechanical strength of a material is distinguished by the nature of the pollutant it contains [35].

### 3.2. TCLP Leaching Test

#### Chemical Properties

The chemical results of S(Cr_2_O_3_, 0.6%, 0.96%, and 1.2%), namely, pH, conductivity, cumulative concentrations, and cumulative leached fractions (C.L.F.) of monitored chemical species (Cr), are presented in Table 3, Table 4 and Table 5.

Based on the findings presented in Table 3, Table 4 and Table 5, two distinct zones can be identified. In zone 1, characterized by pH values lower than 5.44 and time lower than 18 h, there is a significant increase in cumulative chromium concentrations. This phenomenon is primarily attributed to the pH-dependent solubilization of chromium, driven by concentration gradients between the aggressive solution and the pore water within the cementitious matrix. This gradient induces ion transfer through diffusion and the release of chromium present on the outer surface of the mortar monoliths, akin to a water-washing phenomenon [43]. In contrast, in zone 2, where pH levels exceed 5.36 and the time elapsed surpasses 18 h, there is a stabilization in the cumulative chromium concentrations. This stabilization can be attributed to several factors:(i).The dissolution of portlandite occurs when the pH is below 12.5 [44].(ii).The decomposition of C-S-H and ettringite takes place, resulting in a silica gel residue, when pH values are below 10.6 and 8.8, respectively [44].(iii).Strongly soluble phases such as KOH and NaOH dissolve, neutralizing the leachant’s acidity and aligning the pH with that of the strongly basic interstitial solution within the cement mortar. Consequently, chromium release becomes significantly diminished [44].

Furthermore, it is worth noting that the concentrations of stabilized/solidified chromium are substantially lower in comparison to the initial amount of Cr_2_O_3_ introduced into the mixes. This observation highlights that chromium has been effectively retained within the structure of the cement matrices. This confirms the process used for the immobilization of chromium. This shows that the process used in the context of our study is more efficient in terms of the immobilization of Cr compared to the other processes cited in [26]. To better illustrate the influence of Cr_2_O_3_ on the hydration product of cement CEM II, XRD analysis was performed on the samples with Cr_2_O_3_ after 28 days of hardening.

### 3.3. X-ray Diffraction (XRD) Analysis of S/S Mortars

Figure 4a–c and Table 6 show the XRD patterns of the samples, dried at 105 °C for 24 h, crushed, and sieved through a 100 µm sieve.

Based on XRD analyses, it can be seen that in all materials, the SiO_2_ was detected due to presence of the standard sand. Moreover, the appearance of the chromium waste, in all mixes S(Cr_2_O_3_ 0.60, 0.96, and 1.20%), have not blocked the hydration reactions, which is confirmed by the presence of ettringite and portlandite Ca(OH)_2_, which are these reactions’ products [45]. Furthermore, there is a visible increase in the formation of ettringite while using chromium in the mortars mixtures; it is proved by the increased number of ettringite peaks in the XRD patterns [46]. However, it is worth emphasizing that while using chromium in the mortar mixtures, it proves that it is possible to obtain two new crystallized phases, Calcium Chromium Oxide Hydrate CaCrO_4_·2H_2_O and Calcium Chromium Sulfate Hydroxide HydrateCa_6_Cr_2_(SO_4_)_3_(OH)_12_·26H_2_O, that were not detected in previous works summarized in [47]. This is because chromium substitutes the aluminum in ettringite to form a new chromium–ettringite phase.

### 3.4. Infrared Analysis (IR) of S/S Mortars

FTIR spectra of S(Cr_2_O_3_, 0.60%, 0.96%, and 1.20%) mortars are shown in Figure 5, and their vibration bands are presented in Table 7.

The results presented in Table 7 and Figure 5 show that the characteristic bands of Si-O can come from SiO_2_ of the C-S-H and/or the standard sand. We also note the appearance of certain compounds, namely, the metal-oxygen (M-O) band [48], which confirms the formation of bonds between chromium and cement hydrates. The band observed around 1600 cm^−1^ corresponds to intermolecular water [49]. We also note the presence of the C-S-H band, which is the result of hydration reactions. The detection of the band characteristic of portlandite around 3460 cm^−1^ is confirmed by XRD.

### 3.5. Modelling and Simulation

Cumulative leached fractions were modeled using an empirical model developed by Cote et al., and the modeled and simulated values showed remarkable similarity, as demonstrated in Figure 6. The validity tests of the selected models and their parameters are recorded in Table 8.

The parameters recorded in Table 8 reveal that the surface phenomenon prevails over the process. Moreover, the minus sign of K_3_ signifies the delay in leaching, and the infinitely small value of the constant K_4_ evaluates the reaction component, indicating that the phenomena of diffusion and chemical reaction are less significant.

In general, statistical evaluations of the chosen models led to:-A residual variance of the order of zero;-A closeness to one for the coefficients of determination;-The models being integer-validated because the calculated values of the Fisher tests are more significant than the tabulated values of the Fisher tests;-Student’s t-test values demonstrating that all model coefficients are preserved apart from the constant K_3_, which is rejected by the delay in leaching;-The release of chromium being controlled by surface phenomena, as indicated by the simplex method and corroborated by statistical tests.

From the results of the modeling coupling the transfer of chromium by the surface phenomenon, the diffusion, and the dissolution–precipitation reactions, it was observed that the released flows obtained are theoretically very close to those determined experimentally. This good agreement shows that it is possible to apply the present model to predict the release of different heavy metals of previous research that did not take into account the long-term prediction of the release of these toxic pollutants stabilized/solidified and subjected to aggressive environments.

### 3.6. Degradation of Monolithic Cubes during the TCLP Test

To illustrate the degradation from the cumulative concentration released by the monolithic cubes in the medium of study, the criterion of leached rate (percentage) has been used, and is given by the mathematical expression illustrated in the fourth column of Table 9.

Based on the leached rate values, there is no significant degradation in the three S/S mortars. The low leached concentrations of chromium are far below those required by the TCLP test standard; this confirms that chromium is well fixed in the structure of mortars, and consequently chromium becomes non-hazardous. It has also been proved by the results presented in Figure 7. In the presented EDS spectrum of S(Cr_2_O_3_, 1.2%), the presence of Ca, Si, O, Al, Fe, and Al was observed. These elements are the main components of the materials used for the stabilization/solidification technique. It also confirms that the presence of chromium is well fixed in the structure of mortars, and consequently chromium becomes non-hazardous.

Also, the diffractograms of monolithic mortars S(Cr_2_O_3_, 0.60%, 0.96%, and 1.20%) presented in Figure 8 reveal that the intensities of the peaks before and after (core) leaching are of the same order of magnitude. On the other hand, a decrease in peak intensities after leaching is recorded between the core of the matrix and the side exposed to the acidic medium during the TCLP test. This shows a slight washing of the contact surface due to dissolution or precipitation reactions.

The degradation of monolithic cubes during the TCLP test was evaluated from the viewpoint of the intensities of portlandite peaks before and after leaching. This method consists of measuring the average intensity of portlandite peaks before and after leaching. It is used as a criterion for evaluating the degradation of monolithic cubes of mortars [29,36,49].

In accordance with the trends observed in the average intensities of the portlandite peaks, as shown in Figure 8, a slight reduction in the intensity of portlandite was observed when comparing scans BL (Before Leaching), HAL (Heart After Leaching), and EIAF (Exposed Interface After Leaching). This phenomenon can be attributed to the dissolution of portlandite at interfaces exposed to the aggressive leachant.

When comparing different materials, it is evident that S(Cr_2_O_3_, 1.2%) exhibits lower degradation compared to S(Cr_2_O_3_, 0.6%) and S(Cr_2_O_3_, 0.96%). This superior performance can be attributed to its higher mechanical strength in comparison to S(Cr_2_O_3_, 0.6%) and S(Cr_2_O_3_, 0.96%).

Based on the degradation results, we recommend that industrials and policymakers use chromium-rich waste in cementitious materials due to the beneficial contribution of chromium, which reacts with cement hydrates, increases their mechanical strength, and enhances their durability towards aggressive environments.

## 4. Conclusions

For a better understanding of the phenomena that govern the process of stabilization/solidification and the release of chromium in mortars, complex research was conducted. This research includes an evaluation of the chromium’s solidification/stabilization process based on analyses performed using XRD and statistical analyses of the empirical Cote model.

It has been proved that it is possible to transform hazardous waste into non-hazardous material in cementitious composites by their synergistic use. The low leached concentrations of chromium are far below those required by the TCLP test standard. This confirms that chromium is well fixed in the structure of mortars, and consequently chromium becomes non-hazardous, which was confirmed by the results of SEM-EDX. It has also been noticed that the incorporation of chromium-rich waste into the cementitious matrices increases the mechanical strength and improves the durability of these materials. In this context, the use of this type of waste as an addition or partial replacement of cement or aggregate allows reducing the consumption of raw materials (preserves natural resources), reducing pollution, reducing energy use, reducing the volume of waste that must be treated and disposed of, and reducing CO_2_ emissions. It is beneficial to the cement industry, which is one of the most polluting sectors with a contribution of almost 7% of global CO_2_ emissions [50].

The simplicity of implementing the chromium stabilization/solidification process, the availability of hydraulic binders (cement), and their low cost compared to other techniques (namely, vitrification, precipitation of waste by acids, decantation, adsorption, incineration, and stabilization/solidification by geopolymer), high mechanical strength, great ability to chemically immobilize chromium inside the phases, low permeability, and relatively high durability have made the process of stabilization/solidification by cement more reliable and adequate for chromium-rich waste treatment [26].

It was observed that chromium substitutes the aluminum in ettringite to form a new chromium–ettringite phase, forming Calcium Chromium Oxide Hydrate CaCrO_4_·2H_2_O and Calcium Chromium Sulfate Hydroxide Hydrate Ca_6_Cr_2_(SO_4_)_3_(OH)_12_·26H_2_O that results in strengthening the materials.

XRD results explain the success of chromium immobilization within the cement matrices of mortars, in the form of complex phases which were not detected in previous works summarized in [46]. This is because chromium substitutes the aluminum in ettringite to form a new chromium–ettringite phase. The formation of new phases due to the effect of cement hydration and the various interactions between chemical elements considerably reduce the polluting potential of chromium.

Based on the obtained results, we recommend industrials and policymakers use chromium-rich waste in cementitious materials due to the beneficial contribution of chromium, which reacts with cement hydrates, increases their mechanical strength, and enhances their durability towards aggressive environments.

The main limitations of this study are the possibility of using the presented S/S method for materials that are similar to those used during the experimental campaign. Further research might need to be conducted to evaluate comparative analyses of different mixtures of cementitious composites that may include other heavy metals. Performing experimental tests similar to those used in this paper allows comparing the results and evaluating the presented model and its suitability for other heavy metals. Moreover, from the cognitive point of view, it might be beneficial to evaluate the effect of adding non-hazardous waste (such as limestone filler, marble, glass, etc.) to cementitious materials on the release of heavy metals. In addition, it would be advantageous to complete this research work by studying the possibility of using chromium-rich waste as a raw material during the manufacture of cement.

## Figures and Tables

**Figure 1 materials-16-06295-f001:**
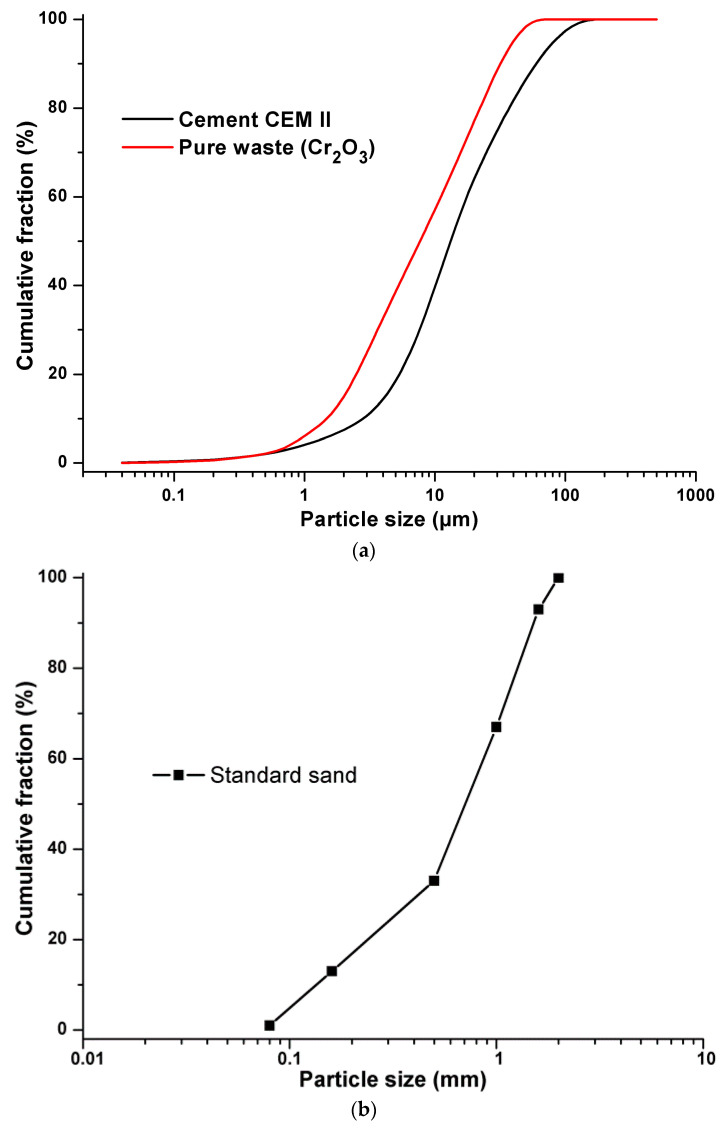
(**a**) Grain size distribution of cement and pure waste. (**b**) Grain size distribution of standard sand.

**Figure 2 materials-16-06295-f002:**
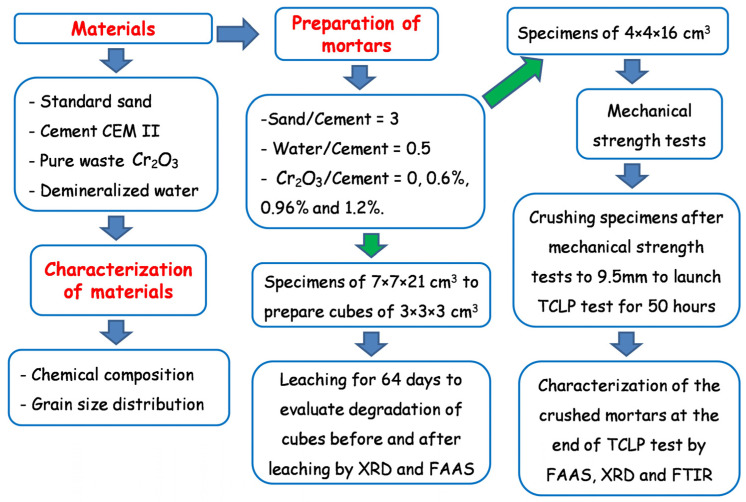
Flowchart of the experimental program undertaken in this study.

**Figure 3 materials-16-06295-f003:**
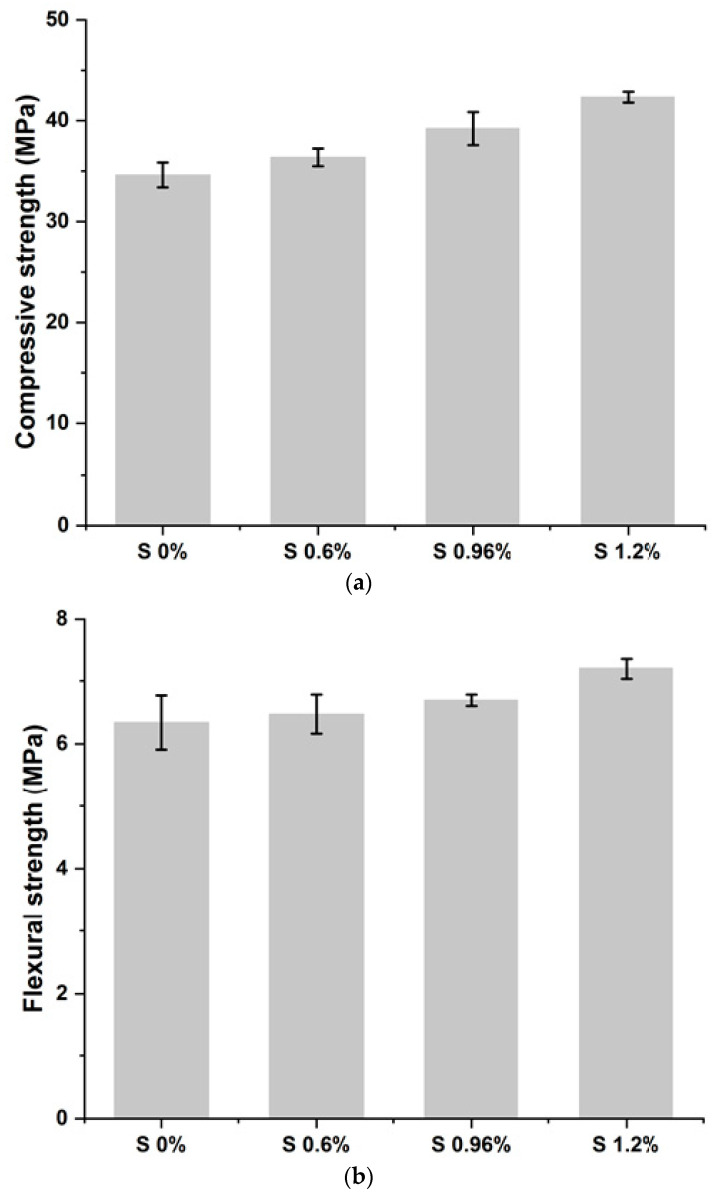
(**a**) Compressive strength of studied mixes. (**b**) Flexural strength of studied mixes.

**Figure 4 materials-16-06295-f004:**
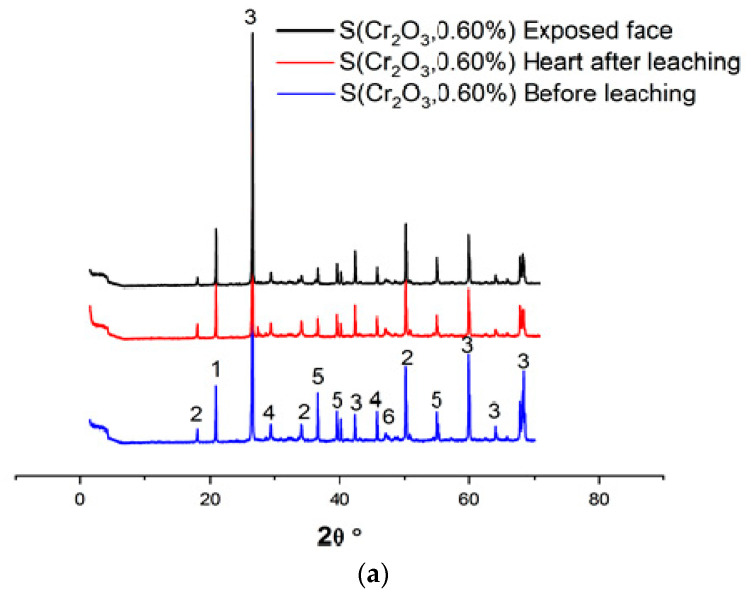
(**a**) XRD patterns of S(Cr_2_O_3_, 0.60%) before and after leaching. (**b**) XRD patterns of S(Cr_2_O_3_, 0.96%) before and after leaching. (**c**) XRD patterns of S(Cr_2_O_3_, 1.20%) before and after leaching.

**Figure 5 materials-16-06295-f005:**
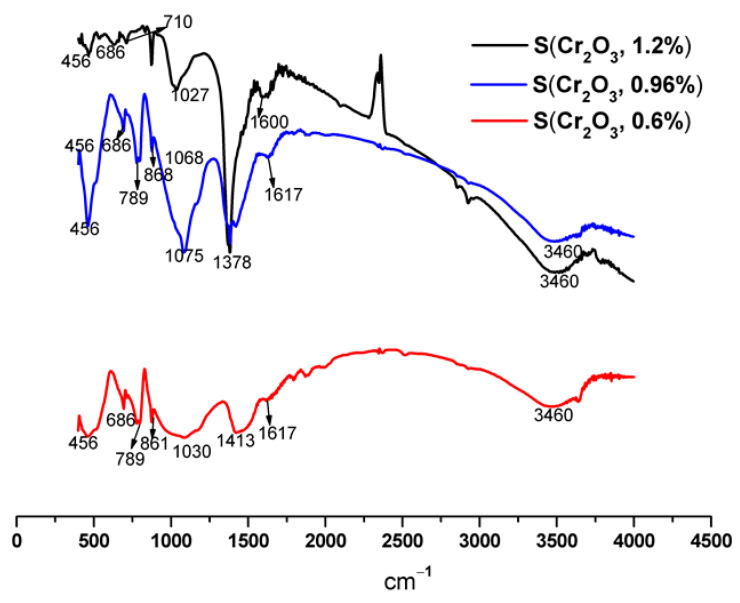
FTIR spectra of S(Cr_2_O_3_, 0.60%, 0.96%, and 1.20%) after 50 h of leaching.

**Figure 6 materials-16-06295-f006:**
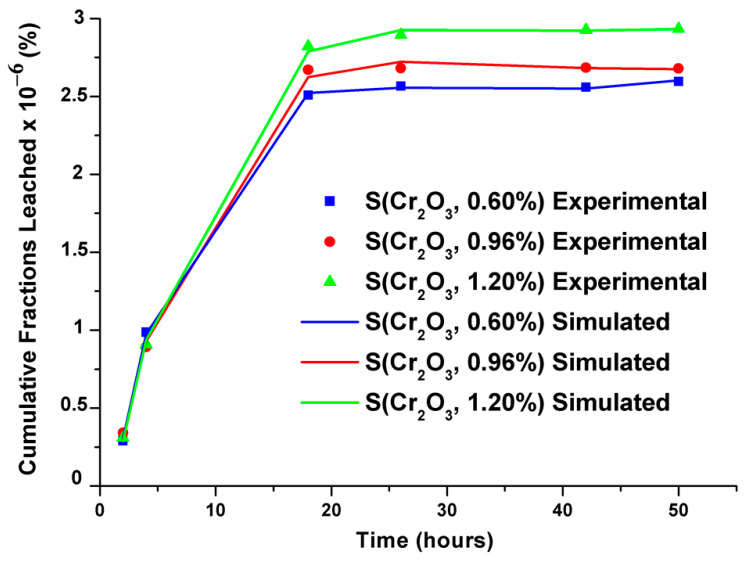
Experimental and simulated cumulative leached fractions of S(Cr_2_O_3_, 0.60%, 0.96%, and 1.20%).

**Figure 7 materials-16-06295-f007:**
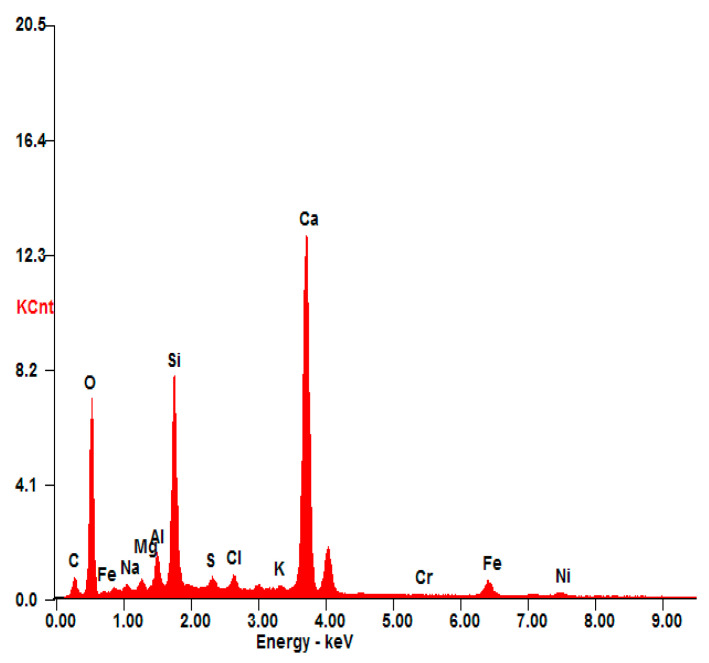
EDS analysis of S(Cr_2_O_3_, 1.20%).

**Figure 8 materials-16-06295-f008:**
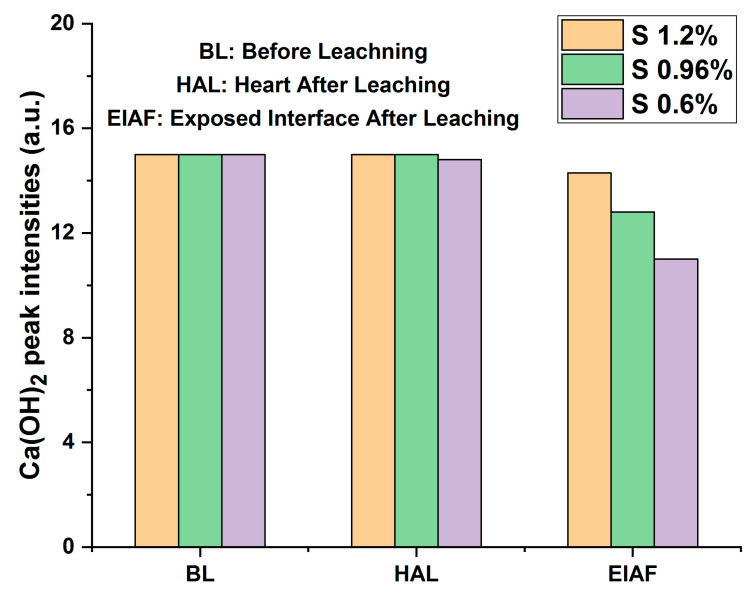
Average peak intensities of portlandite before and after leaching of monolithic cubes of S(Cr_2_O_3_, 0.6%, 0.96%, and 1.2%).

**Table 1 materials-16-06295-t001:** Chemical composition of cement, pure waste, and standard sand.

Compound	SiO_2_	Al_2_O_3_	Fe_2_O_3_	CaO	MgO	SO_3_	K_2_O + Na_2_O	Cr_2_O_3_
Cement (%)	28.36 ± 0.04	6.82 ± 0.09	3.48 ± 0.07	58.57 ± 0.22	1.03 ± 0.02	1.52 ± 0.02	0.17 ± 0.01	n.d.
Pure waste (%)	n.d. ^1^	n.d.	n.d.	n.d.	n.d.	n.d.	n.d.	99.94 ± 0.03
Standard sand (%)	98.05 ± 0.23	0.54 ± 0.05	0.07 ± 0.01	n.d.	n.d.	n.d.	n.d.	n.d.

^1^ n.d. Not detected.

**Table 2 materials-16-06295-t002:** Strength tests on S(Cr_2_O_3_, 0.6%, 0.96%, and 1.2%) and S(Control, 0%) mortars.

Designation	Rc (MPa)	Rf (MPa)
S (Control, 0%)	34.62 ± 1.22	6.34 ± 0.43
S(Cr_2_O_3_, 0.6%)	36.36 ± 0.87	6.47 ± 0.31
S(Cr_2_O_3_, 0.96%)	39.22 ± 1.65	6.69 ± 0.09
S(Cr_2_O_3_, 1.2%)	42.32 ± 0.54	7.20 ± 0.17

**Table 3 materials-16-06295-t003:** Chemical results of S(Cr_2_O_3_, 0.60%) after 50 h of leaching.

Levy	pH	Conductivity (ms)	Cumulative Time (Hours)	[Cr] (μmol/L) in the Leachate	[Cr]_0_(μmol/L) Initial	C.L.F.
1	5.08	4.10	2	0.07499	258,500	2.901 × 10^−7^
2	5.18	4.10	4	0.25509		9.868 × 10^−7^
3	5.38	4.60	18	0.64860		2.509 × 10^−6^
4	5.46	4.60	26	0.66346		2.566 × 10^−6^
5	5.65	5.48	42	0.67115		2.559 × 10^−6^
6	5.80	5.00	50	0.67115		2.596 × 10^−6^

**Table 4 materials-16-06295-t004:** Chemical results of S(Cr_2_O_3_, 0.96%) after 50 h of leaching.

Levy	pH	Conductivity (ms)	Cumulative Time (Hours)	[Cr] (μmol/L) in the Leachate	[Cr]_0_(μmol/L) Initial	C.L.F.
1	5.15	4.30	2	0.13990	411,000	3.404 × 10^−7^
2	5.20	4.30	4	0.36661		8.820 × 10^−7^
3	5.44	5.21	18	1.09753		2.670 × 10^−6^
4	5.53	4.90	26	1.10189		2.681 × 10^−6^
5	5.68	6.18	42	1.10382		2.685 × 10^−6^
6	6.48	5.70	50	1.10164		2.680 × 10^−6^

**Table 5 materials-16-06295-t005:** Chemical results of S(Cr_2_O_3_, 1.20%) after 50 h of leaching.

Levy	pH	Conductivity (ms)	Cumulative Time (Hours)	[Cr] (μmol/L) in the Leachate	[Cr]_0_(μmol/L) Initial	C.L.F.
1	5.19	4.20	2	0.15384	498,000	3.089 × 10^−7^
2	5.17	4.10	4	0.45250		9.086 × 10^−7^
3	5.43	4.70	18	1.40595		2.823 × 10^−6^
4	5.58	4.92	26	1.44230		2.896 × 10^−6^
5	5.73	5.10	42	1.45769		2.927 × 10^−6^
6	5.79	5.29	50	1.46154		2.934 × 10^−6^

**Table 6 materials-16-06295-t006:** Mineralogical compounds detected by X’Pert High Score software (version 2. 1b 2.1.2, 9 January 2005).

Legend	Compound Name	Chemical Formula	Reference Pattern
1	Calcium Chromium Oxide Hydrate	CaCrO_4_·2H_2_O	00-038-1186
2	Portlandite	Ca(OH)_2_	00-004-0733
3	Quartz	SiO_2_	00-046-1045
4	Calcium Chromium Sulfate Hydroxide Hydrate	Ca_6_Cr_2_(SO_4_)_3_(OH)_12_·26H_2_O	00-033-0248
5	Ettringite	Ca_6_Al_2_(SO_4_)_3_(OH)_12_·26H_2_O	00-041-1451
6	Calcite	CaCO_3_	00-005-0586

**Table 7 materials-16-06295-t007:** IR vibration bands of S(Cr_2_O_3_, 0.60%, 0.96%, and 1.20%).

	Wave Number cm^−1^		
	S(Cr_2_O_3_, 0.6%)	S(Cr_2_O_3_, 0.96%)	S(Cr_2_O_3_, 1.2%)
OH of portlandite	3460	3460	3460
H_2_O adsorbed	1617	1617	1600
C-S-H	1378	1378	1378
M-O	1068	1075	1068
C-S-H	861	861	781
Si-O	686	686	622

**Table 8 materials-16-06295-t008:** Simulated parameters and statistical tests of the proposed model for S(Cr_2_O_3_, 0.6%, 0.96%, and 1.2%).

	Surface Phenomenon $K_{1}\left(1-e^{-K_{2}t}\right)$		Diffusion $K_{3}\sqrt{t}$	Chemical Reactions (Dissolution or Precipitation) $K_{4}t$	Correlation Coefficient
Mechanisms Controlling the Release of Cr for S(Cr_2_O_3_, 0.6%)					
Coefficients	K_1_	K_2_	K_3_	K_4_	R^2^
	5.432 × 10^−6^	0.15622	−9.227 × 10^−7^	7.3955 × 10^−8^	0.999
Student’s *t*-test T (n-m, α/2 = 4.514)	51.98	-	29.69	31.16	
Fisher test (F_tabulated_ = 9.78)	11,879				
Residual variance	2.167 × 10^−16^				
Mechanisms controlling the release of Cr for S(Cr_2_O_3_, 0.96%)					
Coefficients	K_1_	K_2_	K_3_	K_4_	R^2^
	5.256 × 10^−6^	0.13359	−7.251 × 10^−7^	5.1065 × 10^−8^	0.999
Student’s *t*-test T (n-m, α/2 = 4.514)	18.32	-	9.16	8.85	
Fisher test (F_tabulated_ = 9.78)	1373				
Residual variance	2.126 × 10^-15^				
Mechanisms controlling the release of Cr for S(Cr_2_O_3_, 1.20%)					
Coefficients	K_1_	K_2_	K_3_	K_4_	R^2^
	5.679 × 10^−6^	0.12961	−7.957 × 10^−7^	5.7839 × 10^−8^	0.999
Student’s *t*-test T (n-m, α/2 = 4.514)	28.01	-	14.47	14.5	
Fisher test (F_tabulated_ = 9.78)	3254				
Residual variance	1.101 × 10^−15^				

**Table 9 materials-16-06295-t009:** Leached rates of chromium during the TCLP test.

Designation	Initial Concentration of Cr (C_0_) en μmol/L	Cumulative Concentration Leached (C_i_) en μmol/L	$\mathrm{Leached}\;\mathrm{Rates}\;(\%)=(C_{i}\times 100)/C_{0}$
S(Cr_2_O_3_, 0.6%)	258,500	0.67	2.59 × 10^−4^
S(Cr_2_O_3_, 0.96%)	411,000	1.10	2.60 × 10^−4^
S(Cr_2_O_3_, 1.2%)	498,000	1.46	2.93 × 10^−4^

## Data Availability

Data are available on request.

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
