# Peer review of "Stabilization of Chromium Waste by Solidification into Cement Composites"

_materials, 2023, doi:10.3390/ma16186295_

Round 1

Reviewer 1 Report

The manuscript titled "Neutralization of chromium waste by solidification into cement composites" presents an important and timely topic. Overall, the manuscript provides valuable insights into a novel method for neutralizing chromium waste. However, as mentioned in the below detailed comments, their first submission could benefit from additional information or clarity to make the manuscript more robust and comprehensive. In addition, the research methodology and findings have been presented in a structured manner. However, before considering it for publication, it would benefit from a language editing to address the punctuation, grammar, and sentence structure issues. Additionally, to enhance its contribution to the field, the authors might consider emphasizing the novelty of their approach more explicitly and contrasting it with existing methods. With these revisions, the manuscript has the potential to offer a valuable contribution to the field and would be suitable for publication in the reputable journal. 

The novelty of the research could be more strongly emphasized, particularly in comparison to existing methods or studies in this area.

The introduction provides a brief context about the significance of chromium waste and its environmental implications. However, it could benefit from a more detailed historical background or a brief literature review of previous attempts to neutralize chromium waste. Please consider the inclusion of the following research:

https://doi.org/10.3390/ijerph15040824

https://doi.org/10.3390/ijerph16071121

https://doi.org/10.3390/min10110932

https://doi.org/10.3390/ma11010141

https://doi.org/10.3390/ijerph18158094

https://doi.org/10.3390/ma16093444

https://doi.org/10.3390/min12050599

https://doi.org/10.3390/ijerph192013666

The authors do state the primary objective of the study, but it might be clearer if the intended outcomes or hypothesis were presented more explicitly.

It would be helpful if the source or grade of the materials, especially the chromium waste, was mentioned.

The step-by-step process is outlined in detail, which is commendable. However, some steps might benefit from visual aids like flowcharts or diagrams for better clarity.

Data Presentation: The results are presented in a structured manner with tables and figures. However, some of the tables could benefit from clearer labeling or legends.

The results are analyzed in the context of the study's objectives. It would be beneficial to highlight unexpected findings or anomalies, if any.

If statistical tools or software were used, they should be mentioned along with the rationale for their choice.

Comparative Analysis: The authors have compared their results with previous studies, which provides context. However, a more in-depth analysis comparing the efficiency of the proposed method with existing methods would enhance the discussion.

Implications: The authors touch upon the environmental implications of their method, but a deeper dive into the long-term impacts or potential commercial applications could add value.

It is essential to mention any limitations of the study, whether they be in terms of the scope, materials used, or potential biases.

The conclusion section might be enriched by also including potential future directions or applications of the research. Also, it would be helpful to include specific recommendations for industries or policymakers.

Some sentences in the manuscript are overly long and complex, making them difficult to comprehend on first reading. Breaking these sentences into simpler constructs will improve clarity and readability. The flow of ideas in the manuscript is mostly logical and sequential. However, there are a few instances where transitions between paragraphs or sections could be smoother. Also, there are instances where simpler or more precise wording could have been chosen to convey the intended meaning more effectively.

Author Response

We would like to thank You for the time and effort while reviewing our Manuscript 

Reviewer 2 Report

This manuscript is framed in the field of circular economy as well as in environmental geochemistry and public health. It is interesting article but it raises many questions that must be corrected or revised. The best way to immobilize toxic and hazardous waste, including nuclear waste, is the application of well-known waste vitrification techniques. The authors omit this fact in the Introduction section. In the title the word "neutralization" is used. The text uses the terms stabilization or immobilization. Authors should clarify the objectives of their work. For example, chromium waste is used for accelerates the hydration process and setting of the mortar or the goal is to "neutralize" a very hazardous waste (Cr VI) ???? . If the goal is the immobilization a very hazardous waste,  I recommend vitrification. For example:

DOI: 10.1016/j.gexplo.2016.07.011

DOI: 10.1016/j.matlet.2016.05.061

The introduction must be rewritten and adapted to the real objectives of the work.

X-ray diffraction analysis indicates that in the cement manufacturing process, certain crystalline phases are formed, but does not demonstrate the presence of Cr VI or Cr III. I imagine that these chrome phases of neoformation benefit the quality of the manufactured material ( It was observed that increasing the amount of chromium in the mixtures  increases the mechanical strength of mortars) or simply reduce the danger for human health. This term should be clarified by the authors. The authors should provide quantitative data on the mineral phases detected by XRD.

Authors write in conclusions section: The TCLP test standard confirms that chromium is well fixed in the structure of mortars and consequently chromium becomes non-hazardous. Authors should provide microphotographs by SEM / EDX that confirm this statement or failing provide bibliographic citations that corroborate it.

In short, the work should be rewritten in some sections and introduce its novelty, clarify the objectives and corroborate some results and statements.

Author Response

(The authors gave the same response as above.)

Round 2

Reviewer 1 Report

The authors have adequately addressed my previous concerns. The manuscript, in its current state, is suitable for publication.

Reviewer 2 Report

Authors have improved their manuscript following all the reviewers comments and recommendations.  Congratulations.